# PSEUDOMETRIC GUIDED ONLINE QUERY AND UPDATE FOR OFFLINE REINFORCEMENT LEARNING

## ABSTRACT

Offline Reinforcement Learning (RL) extracts effective policies from historical data without the need to interact with the environment. However, the learned policy often suffers large generalization errors in the online environment due to the distributional shift. While existing work mostly focuses on learning a generalizable policy, we propose to adapt the learned policy to fit the online environment with limited queries. The goals include querying reasonable actions with limited chances and efficiently modifying the policy. Our insight is to unify these two goals via a proper pseudometric. Intuitively, the metric can compare online and offline states to infer optimal query actions. Additionally, efficient policy updates require good knowledge of the similarity between query results and historical data. Therefore, we propose a unified framework, denoted Pseudometric Guided Offline-to-Online RL ($\text{PGO}^2$). Specifically, in deep Q learning, $\text{PGO}^2$ has a structural design between the Q-neural network and the Siamese network, which guarantees simultaneous Q-network updating and pseudometric learning, promoting Q-network fine-tuning. In the inference phase, $\text{PGO}^2$ solves convex optimizations to identify optimal query actions. We also show that $\text{PGO}^2$ training converges to the so-called bisimulation metric with strong theoretical guarantees. Finally, we demonstrate the superiority of $\text{PGO}^2$ on diversified datasets.

## 1 INTRODUCTION

Offline Reinforcement Learning (RL) leverages large historical data to learn the behavior policy, which seeks the optimality for sequential decision-making without any costs from interacting with the environments (Lange et al., 2012; Levine et al., 2020). This promising feature significantly promotes real-world RL applications, especially when explorative actions are costly. Thus, advances for offline RL have been made in robotic control (Lee et al., 2022), healthcare (Tang & Wiens, 2021), dialogue model (Jaques et al., 2020), recommendation (Xiao & Wang, 2021), and E-commerce (Zhang et al., 2021), etc.

Despite the achievement, the learned policy in offline RL may still suffer large extrapolation errors in online implementations (Dadashi et al., 2021; Fujimoto et al., 2019; Lee et al., 2022). The central reason is the data distributional shift between the offline and online environments. To address this issue, most solutions improve the algorithmic model, like adding constraints in the learning procedure (Fujimoto et al., 2019; Wang et al., 2020), designing ensemble models (Agarwal et al., 2020; Lee et al., 2022; An et al., 2021), or providing better value function estimation (Kumar et al., 2020; Dadashi et al., 2021; Rezaeifar et al., 2022). While these methods provide appealing results, further improvements can be made when the agent can obtain online data. A classic setting is off-policy RL (Levine et al., 2020; Munos et al., 2016) where a data buffer continuously memorizes online data to update the policy.

In many cases, it is often unacceptable to obtain large online samples based on the offline policy or some random explorations. This is because the poor generalization of offline policy may cause high costs for online implementations, let alone random explorations. Therefore, we propose to actively produce query actions to the environment for online states. Subsequently, the environment can evaluate the query and provide feedback (i.e., rewards), which facilitates adapting the offline policy. The general goal is to utilize limited queries to achieve efficient policy adaptation.

More specifically, one needs to (1) produce proper query actions and (2) accurately modify the offline policy according to limited query results. Basically, the state distributional shift challenges

goal (1). In addition, limited query results challenge goal (2). For example, for deep Q-learning, it is hard to adjust a huge parameter set in the Q-neural network with few samples. To address these issues, one principled approach is to find a proper similarity measure for the state-action pairs in a Markov Decision Process (MDP). Then, for goal (1), we can infer query actions based on similar states and actions in the historical dataset. For goal (2), the differences between each online query with all the historical samples provide rich information that can be intelligently employed to update the Q-network. In general, we demand a good pseudometric (Dadashi et al., 2021) to link offline datasets with online queries.

In particular, the well-defined pseudometric should guarantee that similar states must have a small difference between the expected rewards. In an MDP, the above definition is known as the bisimulation metric (Ferns & Precup, 2014; Dadashi et al., 2021). To approximate the bisimulation metric, (Dadashi et al., 2021) has developed a pseudometric learning model that takes in the historical data and outputs the pseudometric, a modified bisimulation metric in the state-action space. Although this model benefits goal (1), goal (2) is still unreachable since the pseudometric learning and the Q-network updating are decoupled.

Therefore, we propose $PGO^2$: Pseudometric Guided Offline-to-Online RL. $PGO^2$ is a unified framework to learn the pseudometric, update the Q-network, and infer query actions. Specifically, $PGO^2$ employs Siamese networks to learn the similarity of the state-action pairs (Bromley et al., 1993; Dadashi et al., 2021). More importantly, we restrict the Q-network and the Siamese network to sharing parameters. Consequently, learning the pseudometric between each query result and all historical data updates the parameters of the Q-network. This updating scheme provides sufficient evaluations from limited query results, which enables the offline policy to quickly adapt to the online environment. Moreover, the Q-network in $PGO^2$ is designated as Partially Input Convex Neural Network (PICNN) (Amos et al., 2017a). Thus, the inference of the action query has optimality by solving convex optimizations. For theoretical guarantees, we also show the theorems of convergence to the bisimulation metric and the global optimality for the query process. Finally, we observe significant improvements of $PGO^2$ in multiple RL tasks under the offline-to-online setting.

## 2 RELATED WORK

**Offline RL**. In the Introduction, we have presented a review of different models to optimize the sequential decisions using offline data. In addition, some other works view the offline RL as a supervised learning model to generate sequences. (Janner et al., 2021) employs a trajectory Transformer to predict the sequence of states, actions, and rewards. (Chen et al., 2021) also builds these sequences by a masked Transformer. Finally, (Schweighofer et al., 2021; Sinha et al., 2022) study how the historical data can impact the learned policy in offline RL. However, these studies do not provide a reasonable approach for online queries and updates.

**Online Implementations to Improve Offline RL**. We categorize existing studies into the following groups. The first group requires the historical data to be informative so that they can abstract prior knowledge, e.g., skills (Pertsch et al., 2020), primitive behaviors (Ajay et al., 2020), and behavioral priors (Singh et al., 2020), for online implementations. However, we may not have enough high-quality offline data. The second group admits imperfect data and employs online data to update the learned policy. (Nair et al., 2020) proposes to restrict Kullback–Leibler (KL) divergence of the policy for both offline and online learning, which requires a certain amount of online data. (Lee et al., 2022) weighs the offline and online data based on the density ratio so that the offline Q-network can be fine-tuned. Similarly, enough online samples are required to accurately estimate the density ratio. We make significant contributions by showing how a generalized pseudometric can guide the online action query and policy update, even with limited query opportunities.

**State-action Similarity Metric for MDPs**. The bisimulation metric uses rewards to determine the similarity of two states and/or actions (Ferns & Precup, 2014). Another similarity definition is the MDP homomorphism by considering both the reward and the transition probability (Ravindran & Barto, 2003). Many studies have been done to approximate these metrics (Castro, 2020; Dadashi et al., 2021; van der Pol et al., 2020). Our pseudometric learning is similar to (Dadashi et al., 2021) to approximate the bisimulation metric in the state-action space with theoretical guarantees. However, we have a unique structural design with convex optimizations to infer optimal queries and conduct sufficient policy updating.

## 3 METHODS

### 3.1 BACKGROUND AND PROBLEM FORMULATION

An MDP can be formalized as a tuple $(\mathcal{S}, \mathcal{A}, r, P, \gamma)$, where $\mathcal{S}$ is the state space, $\mathcal{A}$ is the action space, $r : \mathcal{S} \times \mathcal{A} \times \mathcal{S} \to \mathbb{R}$ is the reward function, $P(\boldsymbol{s}_{k+1}|\boldsymbol{s}_k, \boldsymbol{a}_k)$ is the state transition function to measure the probability from $\boldsymbol{s}_k$ to $\boldsymbol{s}_{k+1}$ with a given action $\boldsymbol{a}_k$, and $\gamma \in [0, 1)$ is a discount factor. The agent can follow a policy $\pi : \mathcal{S} \to \mathcal{A}$ to determine an action for a given state. In the optimization of RL, the general goal is to find an optimal policy to maximize $\mathbb{E}_\pi \big[ \sum_{k=0}^\infty \gamma^k r(\boldsymbol{s}_k, \boldsymbol{a}_k, \boldsymbol{s}_{k+1}) \big]$.

Q-learning can solve the optimization in an iterative manner. For each policy $\pi$, the Q-function is a value function for the state-action pair: $Q^\pi(\boldsymbol{s}, \boldsymbol{a}) = \mathbb{E} \big[ \sum_{k=0}^\infty \gamma^k r(\boldsymbol{s}_k, \boldsymbol{a}_k, \boldsymbol{s}_{k+1}|\boldsymbol{s}_0 = \boldsymbol{s}, \boldsymbol{a}_0 = \boldsymbol{a}) \big]$. Then, the optimal policy is to solve the optimization $\max_\pi Q^\pi(\boldsymbol{s}, \boldsymbol{a})$ to obtain the policy function for all states. With advances in deep learning, neural networks are used to approximate the Q-function, e.g., Deep Q-network (Mnih et al., 2013). To obtain the optimal policy, one can iteratively update the Q-network:

$$Q_{t+1}(\boldsymbol{s}_k, \boldsymbol{a}_k) = Q_t(\boldsymbol{s}_k, \boldsymbol{a}_k) + \alpha \big( r(\boldsymbol{s}_k, \boldsymbol{a}_k) + \gamma \max_{\boldsymbol{a}} Q_t(\boldsymbol{s}_{k+1}, \boldsymbol{a}) - Q_t(\boldsymbol{s}_k, \boldsymbol{a}_k) \big), \quad (1)$$

where $t$ and $k$ are the indices for the episode and state-action pair, respectively. $\alpha$ is the learning rate. This updating can happen both offline and online. For example, in the offline RL, we can compute equation 1 using state transition sample $(\boldsymbol{s}_k, \boldsymbol{a}_k, \boldsymbol{s}_{k+1}, r_k) \in \mathcal{D}$, where we denote $r_k = r(\boldsymbol{s}_k, \boldsymbol{a}_k, \boldsymbol{s}_{k+1})$ and the set of $N$ samples as $\mathcal{D} = \big\{ (\boldsymbol{s}_i, \boldsymbol{a}_i, \boldsymbol{s}_{i+1}, r_i) \big\}_{i=1}^N$.

In this paper, we focus on offline training and online queries with a start state $\tilde{\boldsymbol{s}}_1$ that may be dissimilar to the states in $\mathcal{D}$, e.g., an unseen block in the grid world (Crook & Hayes, 2003). We add the notation $\tilde{\phantom{x}}$ to indicate online data and formally define the problem as follows.

- Input: historical dataset $\mathcal{D}$ and an online start state $\tilde{\boldsymbol{s}}_1$.
- Query Task: use $\mathcal{D}$ to learn an offline Q-network $Q(\cdot)$. Then, use $\mathcal{D}$, $Q(\cdot)$, and $\tilde{\boldsymbol{s}}_1$ to generate sequential and optimal query actions $\{\tilde{\boldsymbol{a}}_1, \cdots, \tilde{\boldsymbol{a}}_M\}$, where $M$ is a small integer. Note that repetitions can happen for the query action sequence.
- Update Task: use $\mathcal{D}$ and the query results $\tilde{\mathcal{D}} = \big\{ (\tilde{\boldsymbol{s}}_i, \tilde{\boldsymbol{a}}_i, \tilde{\boldsymbol{s}}_{i+1}, \tilde{r}_i) \big\}_{i=1}^M$ to update the offline $Q(\cdot)$ and produce an online $\tilde{Q}(\cdot)$.
- Output: $\tilde{\mathcal{D}}$ and $\tilde{Q}(\cdot)$ for online implementations.

We will illustrate how to use limited resources to achieve optimal queries and updates, shown in the left part of Fig. 1. We term the framework as Pseudometric Guided Offline-to-Online RL (PGO$^2$).

### 3.2 PGO$^2$ ARCHITECTURE AND OFFLINE TRAINING

(Dadashi et al., 2021) proposes a pesudometric learning framework to approximate the modified bisimulation metric (i.e., a pseudometric) in the state-action space. Specifically, they introduce a Siamese neural network $\Phi$ to learn the pesudometric $d_\Phi(\boldsymbol{s}_i, \boldsymbol{a}_i; \boldsymbol{s}_j, \boldsymbol{a}_j) = \big|\big|\Phi(\boldsymbol{s}_i, \boldsymbol{a}_i) - \Phi(\boldsymbol{s}_j, \boldsymbol{a}_j)\big|\big|$, where $||\cdot||$ is the Euclidean distance. By the definition of the proposed bisimulation metric in (Dadashi et al., 2021) (see also in Section 4), they design another Siamese network $\Psi$ to compute the bootstrapped estimate of the state bisimulation metric. This metric measures the similarity between states with randomly selected actions from $\mathcal{U}(\mathcal{A})$, i.e., a uniform distribution over the action space. Then, they propose a concurrent learning scheme to optimize the following loss functions:

$$L_\Phi = \mathbb{E}\bigg( \big|\big|\Phi(\boldsymbol{s}_i, \boldsymbol{a}_i) - \Phi(\boldsymbol{s}_j, \boldsymbol{a}_j)\big|\big| - |r_i - r_j| - \gamma\big|\big|\Psi(\boldsymbol{s}_{i+1}) - \Psi(\boldsymbol{s}_{j+1})\big|\big| \bigg)^2,$$

$$L_\Psi = \mathbb{E}\bigg( \big|\big|\Psi(\boldsymbol{s}_i) - \Psi(\boldsymbol{s}_j)\big|\big| - \frac{1}{n} \sum_{\boldsymbol{u}_k \sim \mathcal{U}(\mathcal{A}), k=1}^n \big|\big|\Phi(\boldsymbol{s}_i, \boldsymbol{u}_k) - \Phi(\boldsymbol{s}_j, \boldsymbol{u}_k)\big|\big| \bigg)^2, \quad (2)$$

where $(\boldsymbol{s}_i, \boldsymbol{a}_i, r_i, \boldsymbol{s}_{i+1}) \sim \mathcal{D}$, $(\boldsymbol{s}_j, \boldsymbol{a}_j, r_j, \boldsymbol{s}_{j+1}) \sim \mathcal{D}$. The pseudometric $d_\Phi$ is trained by $\mathcal{D}$, which does not require Q-network for the offline RL.

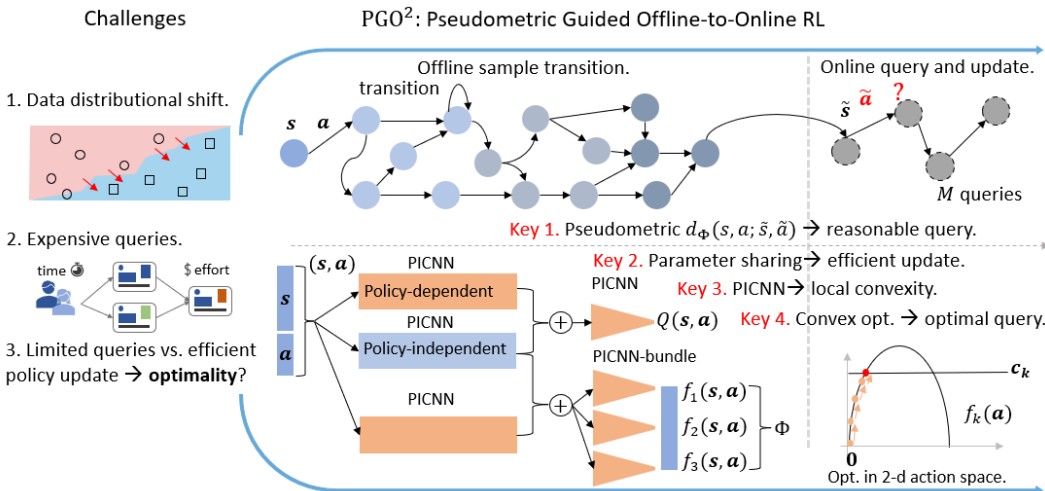

Figure 1: The proposed framework.

We improve the above learning process by coupling the pseudometric learning and the Q-network training in a parameter-sharing manner. The intuition is that $Q(s, a)$ and $\Phi(s, a)$ both serve as evaluations of the state-action pair $(s, a)$, although $Q(s, a)$ is the value function and $\Phi(s, a)$ is an feature embedding related to the reward differences. Therefore, we restrict the $Q(\cdot)$ and $\Phi(\cdot)$ to share some common features. In general, parameter-sharing enjoys two benefits. First, training each network improves the convergence of the other network as they have similar outputs. Second, updating the $\Phi$ network is cheap in the online mode. For example, one online query result can help to largely re-train the $\Phi$ network by comparing the query sample with all samples in $\mathcal{D}$.

To elaborate on the architecture of the two networks, we note that their output differences are due to the different policy distributions. $Q(\cdot)$ considers the policy by the learned Q-function while $\Phi(\cdot)$ considers a uniform distribution over the action space. Thus, we consider *policy-dependent* and *policy-independent* features for both networks. The policy-independent features are common to evaluate the state-action pair no matter what the subsequent policy is taken, e.g., the instant reward. Thus, we make these features common with shared parameters, as shown in the blue block at the bottom of Fig. 1. Correspondingly, the policy-dependent features have different parameters, as shown in the orange blocks at the bottom of Fig. 1. Finally, these two features are summed to jointly output the final values. The summation is due to the weighted summation formula to evaluate the state-action pairs, e.g., equation 1.

In the offline training of PGO$^2$, due to the parameter-sharing design, $\Phi(\cdot)$, $\Psi(\cdot)$, and $Q(\cdot)$ should be trained concurrently to guarantee efficient updating. We demonstrate the specific training process as follows. First, in each iteration, the three networks are updated via random samples from $\mathcal{D}$, like experience replay (Mnih et al., 2015) for off-policy RL. Second, with the selected samples, parameters of $\Phi(\cdot)$ and $\Psi(\cdot)$ can be directly trained by minimizing the loss in equation 2. Third, we train $Q(\cdot)$ according to the temporal difference in equation 1. In particular, we need to solve the optimization $\max_a Q_t(s_{k+1}, a)$ to seek the optimal actions for $s_{k+1}$.

In vanilla Deep Q-network (Mnih et al., 2013), the optimization can be easily solved since the output of the Q-network is the action-Q pairs, which helps to do maximization by comparison. However, in our PGO$^2$, the action is treated as the input for the $Q(\cdot)$. Thus, $\max_a Q_t(s_{k+1}, a)$ is the optimization with respect to the input of a neural network. We employ the Partially Input Convex Neural Network (PICNN) (Amos et al., 2017b) to model $-Q(s, a)$. Specifically, with non-negative weight and Rectified Linear Unit (ReLU) activations, we can make $-Q(s, a)$, a PICNN, to be convex in $a$. Then, we can utilize the gradient-based method or the Projected Newton method (Bertsekas, 1982) to solve the convex optimization $\max_a Q_t(s_{k+1}, a)$.

The above optimization is an inference problem for PICNN (Amos et al., 2017b). Similarly, querying an action online is another inference process. Specifically, to seek optimal query action, we compare the online state with state-action pairs in $\mathcal{D}$ to find the optimal query action. Thus, the inference is related to not only $Q(\cdot)$ but also $\Phi(\cdot)$. To guarantee to find the optimal query action, we

propose a bundle of PICNNs to model $-\Phi(\cdot)$ such that each output neuron of $-\Phi(s, a)$ is convex in input action $a$, shown in the bottom part of Fig. 1. In Section 3.3, we show the PICNN-bundle brings convex optimization for online inference.

---

**Algorithm 1** PGO$^2$ Offline Training

---

**Input:** training dataset $\mathcal{D}$.
**Initialize:** discount factor $\gamma \in [0, 1)$, maximum episode $T$, PICNN for Q-network $-Q(s, a)$, target network $Q^{'}(\cdot) = Q(\cdot)$, PICNN bundle for $-\Phi(s, a)$, a neural network for $\Psi(a)$, and update interval $T_0$ for $Q^{'}(\cdot), \Phi(\cdot)$, and $\Psi(\cdot)$.
**while** $t \leq T$ **do**
    Sample a random minibatch $\mathcal{B} \subset \mathcal{D}$
    **for** $(s_m, a_m, r_m, s_{m+1}) \in \mathcal{B}$ **do**                 ▷ Experience replay.
        Solve the optimization $\max_a Q^{'}(s_{m+1}, a)$ to obtain $a_{m+1}$.
        $y_m = r_m + \gamma Q^{'}(s_{m+1}, a_{m+1})$.
    Train $Q(\cdot)$ using training data $\{s_{m+1}, a_{m+1}, y_m\}_m$, where $\{s_{m+1}, a_{m+1}\}_m$ are the input and $\{y_m\}_m$ are the output.
    Stack the minibatch $\mathcal{B}_1 = \{\mathcal{B}_1, \mathcal{B}\}$.
    **if** $t \mod T_0 = 0$ **then**
        $Q^{'}(\cdot) = Q(\cdot)$                       ▷ Update $Q^{'}(\cdot)$.
        Use $(s_m, a_m, r_m, s_{m+1}) \in \mathcal{B}_1$ to train $\Phi(\cdot)$ and $\Psi(\cdot)$ by solving the optimization in equation 2.                                      ▷ Update $\Phi(\cdot)$ and $\Psi(\cdot)$.
**Output:** trained neural networks $Q(\cdot), \Phi(\cdot)$, and $\Psi(\cdot)$.

---

Finally, training instability, a common issue for Deep Q-Learning, also exists in our training process. In particular, one needs to guarantee a stable policy for several training episodes to improve convergence. In PGO$^2$, the policy instability comes from the continuous update of both $Q(\cdot)$ and $\Phi(\cdot)$ with common parameters. To address this issue, we follow (Fan et al., 2020) and employ a target Q-network that is a copy of the Q-network but updated occasionally. This guarantees a stable policy before the target Q-network is updated. Similarly, we restrict $\Phi(\cdot)$ and $\Psi(\cdot)$ to be updated as frequently as the target Q-network to achieve a stable policy. In general, we propose the offline training algorithm for PGO$^2$ in Algorithm 1.

## 3.3 PGO$^2$ ONLINE INFERENCE FOR OPTIMAL QUERIES

In an online mode, PGO$^2$ needs to infer a reasonable query action for a given online state $\tilde{s}_1$. Since $\tilde{s}_1$ may be an Out-Of-Distribution (OOD) sample, one cannot directly infer the query action by $Q(\cdot)$. Consequently, our strategy is to find the closest state-action pair $(s_c, a_c)$ in $\mathcal{D}$ based on the learned metric $d_\Phi$. By Dadashi et al. (2021), learning $d_\Phi$ has good convergence to the unique fixed point pseudometric to measure the bisimulation of state-action pairs in MDPs. Thus, it is reasonable to utilize $d_\Phi$ as an optimization objective to infer good online actions. Specifically, we propose the following optimization:

$$a_c = \arg\max_a Q(s_c, a), \quad \tilde{a}_c = \arg\min_{(s_c, a_c) \in \mathcal{D}, \tilde{a}} d_\Phi(s_c, a_c; \tilde{s}_1, \tilde{a}), \tag{3}$$

where $\tilde{a}_c$ is our target query action. The first optimization in equation 3 can be easily solved as convex optimization. Then, we can input each $(s_c, a_c)$ pairs and the online initial state $\tilde{s}_1$ to $d_\Phi(\cdot; \cdot)$ to solve the second optimization. With the PICNN-bundle to model $\Phi(\cdot)$, We prove the second optimization is also convex for each state-action pair in $\mathcal{D}$ in Proposition 2. Then, we can also utilize the gradient-based method to solve the optimization and find $\tilde{a}_c$. For example, the bottom right of Fig. 1 visualizes the gradient-based optimization in the 2-dimensional space. After the optimization, we set $\tilde{a}_1 = \tilde{a}_c$ as the first query action for $\tilde{s}_1$.

After performing the query action $\tilde{a}_1$, the agent will move to $\tilde{s}_2$. The online environment will evaluate $(\tilde{s}_1, \tilde{a}_1, \tilde{s}_2)$ and produce the reward $\tilde{r}_1$. The online sample $(\tilde{s}_1, \tilde{a}_1, \tilde{s}_2, \tilde{r}_1)$ can facilitate to update $\Phi(\cdot), \Psi(\cdot)$, and $Q(\cdot)$, discussed in Section 3.4. Based on the updated networks and $\tilde{s}_2$, we can continue the optimization in equation 3 and produce $\tilde{a}_2$. Iteratively, we can obtain the query dataset $\tilde{\mathcal{D}} = \left\{ \left(\tilde{s}_i, \tilde{a}_i, \tilde{s}_{i+1}, \tilde{r}_i\right) \right\}_{i=1}^{M}$, accomplishing the Query Task.

### 3.4 PGO$^2$ ONLINE TRAINING FOR EFFICIENT POLICY UPDATES

To efficiently update the Q-network, we start from the decomposed architecture in Fig. 1. Specifically, the policy-independent features play a key role to evaluate the instant online rewards that may be completely different from the offline rewards. Therefore, we need to significantly and frequently update the corresponding parameters. Next, the update is cheap as one can train one query result versus all samples in $\mathcal{D}$, bringing accurate and stable function approximation. Thus, we update the related parameters each time after an online query.

Basically, we only enable the parameters $\Phi(\cdot)$ and $\Psi(\cdot)$ trainable, which include the policy-independent parameters for $Q(\cdot)$ due to the parameter-sharing. Then, we retrain PGO$^2$ by using the query result (e.g., $(\tilde{s}_i, \tilde{a}_i, \tilde{s}_{i+1}, \tilde{r}_i)$ for the $i^{th}$ query) versus all the samples in $\mathcal{D}$, which is beneficial for the next action query.

The parameters of policy-dependent features for $Q(\cdot)$, however, should not be frequently updated to maintain stability. Thus, after obtaining $\tilde{\mathcal{D}}$, we utilize all samples in $\tilde{\mathcal{D}}$ to retrain the parameters for policy-dependent features. This fine-tunes $Q(\cdot)$ to make it applicable for online implementations. We summarize the online algorithm for PGO$^2$ in Algorithm 2, which ends the Update Task.

---

**Algorithm 2** PGO$^2$ Online Inference and Update

---

    **Input:** training dataset $\mathcal{D}$, trained neural networks $Q(\cdot)$, $\Phi(\cdot)$, and $\Psi(\cdot)$, and an online start state $\tilde{s}_1$.
    **Initialize:** maximum number of query actions $M$.
    **while** $m \leq M$ **do**
        For online state $\tilde{s}_m$, use gradient-based method to solve the convex optimization in equation 3 and obtain $\tilde{a}_m$.
        Perform the online action $\tilde{a}_m$ and obtain the reward $\tilde{r}_m$ and the next state $\tilde{s}_{m+1}$.
        Retrain $\Phi(\cdot)$ and $\Psi(\cdot)$ in PGO$^2$ by $(\tilde{s}_m, \tilde{a}_m, \tilde{s}_{m+1}, \tilde{r}_m)$ and all the samples in $\mathcal{D}$.
        $\tilde{s}_m = \tilde{s}_{m+1}$.
        $\tilde{\mathcal{D}} = \{\tilde{\mathcal{D}}, (\tilde{s}_m, \tilde{a}_m, \tilde{s}_{m+1}, \tilde{r}_m)\}$
    Enable the policy-dependent parameters trainable in $Q(\cdot)$. Retrain $Q(\cdot)$ using all the data in $\tilde{\mathcal{D}}$.
    **Output:** trained neural networks $\tilde{Q}(\cdot)$, $\tilde{\Phi}(\cdot)$, and $\tilde{\Psi}(\cdot)$ for online environments.

---

## 4 THEORETICAL ANALYSIS

In this section, we provide theoretical support for pseudometric learning and online inference. First, Dadashi et al. (2021) provides that iterative calculations from samples can converge to the fixed point of the pseudometric. Let $d(\cdot; \cdot)$ be a pseudometric defined over the state-action space and $\mathcal{F}$ be an operator for the pseudometric such that:

$$\mathcal{F}(d)(s_1, a_1; s_2, a_2) = \begin{cases} |r_i - r_j| + \gamma \mathbb{E}_{u \sim \mathcal{U}(\mathcal{A})} d(s_{i+1}u; s_{j+1}, u) & \text{if } s_1, a_1, s_2, a_2 = s_i, a_i, s_j, a_j \\ d(s_1, a_1; s_2, a_2) & \text{otherwise,} \end{cases}$$

(4)

where $(s_i, a_i, s_{i+1}, r_i)$ and $(s_j, a_j, s_{j+1}, r_j)$ are samples from $\mathcal{D}$. Then, Dadashi et al. (2021) introduces the following proposition to show the convergence.

**Proposition 1.** *Suppose sufficient coverage of the state-action space: $\exists \epsilon > 0$ such that for any state-action pairs $(s_i, a_i), (s_j, a_j) \in (\mathcal{S} \times \mathcal{A}) \times (\mathcal{S} \times \mathcal{A})$, $(s_i, a_i), (s_j, a_j)$ are sampled with at least probability $\epsilon$, then the repeated application of $\mathcal{F}$ converges to the fixed point $d^*$, i.e., the bisimulation metric over the state-action space.*

Proposition 1 guarantees to learn a good pseudometric in the iterative manner in Section 3.2, an approximate version of the operator $\mathcal{F}$. Notably, our pseudometric learning is consistent between offline and online environments. Thus, if the offline data samples are not sufficient as required in Proposition 1, the online queries further improve the convergence. Secondly, we show that our query mechanism can produce optimal query action based on our PICNN-bundle model for $\Phi(\cdot)$.

**Proposition 2.** *Assume $Q(\cdot)$, $\Phi(\cdot)$, and $d_\Phi(\cdot; \cdot)$ are defined in Section 3.2. Let $s_i$ be an arbitrary state from $\mathcal{D}$, $a_i = \arg\max_a Q(s_i, a)$, and $\tilde{s}_j$ be an online state. Then, the optimization $\min_{\tilde{a}} d_\Phi(s_i, a_i; \tilde{s}_j, \tilde{a})$ is convex in $\tilde{a} \in \tilde{\mathcal{A}}$, where $\tilde{\mathcal{A}}$ is defined in the proof.*

*Proof.* In the optimization $\min_{\tilde{\boldsymbol{a}}} d_\Phi(\boldsymbol{s}_i, \boldsymbol{a}_i; \tilde{\boldsymbol{s}}_j, \tilde{\boldsymbol{a}})$, the inputs $\boldsymbol{s}_i$, $\boldsymbol{a}_i$, and $\tilde{\boldsymbol{s}}_j$ are fixed to the Siamese neural network. Then, we simplify the objective as $L(\tilde{\boldsymbol{a}}) = \sum_k^K \left( f_k(\tilde{\boldsymbol{a}}) - c_k \right)^2$, where $K$ is the number of dimension for $\Phi(\cdot)$, $f_k$ is output the $k^{th}$ entry in $\Phi(\tilde{\boldsymbol{s}}_j, \tilde{\boldsymbol{a}})$, and $c_k$ is the $k^{th}$ constant entry in $\Phi(\boldsymbol{s}_i, \boldsymbol{a}_i)$. Let $\boldsymbol{h}$ be a perturbation direction of $\tilde{\boldsymbol{a}}$ and $t$ be a small step size. Then, we can calculate the second-order derivative based on the chain rule:

$$\frac{d^2}{dt^2}\big|_{t=0} L(\tilde{\boldsymbol{a}} + t\boldsymbol{h}) = \frac{d}{dt}\big|_{t=0} 2 \sum_k^K \left( f_k(\tilde{\boldsymbol{a}} + t\boldsymbol{h}) - c_k \right) \frac{d}{dt} f_k(\tilde{\boldsymbol{a}} + t\boldsymbol{h}),$$

$$= 2 \sum_k^K \left( \frac{d}{dt}\big|_{t=0} f_k(\tilde{\boldsymbol{a}} + t\boldsymbol{h}) \right)^2 + \left( f_k(\tilde{\boldsymbol{a}}) - c_k \right) \frac{d^2}{dt^2}\big|_{t=0} f_k(\tilde{\boldsymbol{a}} + t\boldsymbol{h}).$$

Then, we define $\tilde{\mathcal{A}} = \{\tilde{\boldsymbol{a}} | \forall 1 \leq k \leq K, f_k(\tilde{\boldsymbol{a}}) \leq c_k\}$. Since we utilize PICNN to model $-f_k(\cdot), \forall 1 \leq k \leq K$, the concavity leads to $\frac{d^2}{dt^2}\big|_{t=0} f_k(\tilde{\boldsymbol{a}} + t\boldsymbol{h}) < 0$. Thus, $\forall \tilde{\boldsymbol{a}} \in \tilde{\mathcal{A}}$, $\frac{d^2}{dt^2}\big|_{t=0} L(\tilde{\boldsymbol{a}} + t\boldsymbol{h}) > 0$ and the convexity is proved. □

The global optimal solution to minimize $L(\tilde{\boldsymbol{a}})$ lies in the boundary of $\tilde{\mathcal{A}}$, i.e., $\partial\tilde{\mathcal{A}} = \{\tilde{\boldsymbol{a}} | \forall 1 \leq k \leq K, f_k(\tilde{\boldsymbol{a}}) = c_k\}$. Although it is hard to quantify the region of $\tilde{\mathcal{A}}$ in Proposition 2, we can initialize $\tilde{\boldsymbol{a}}_0 = \boldsymbol{0}$. The reason is that we can easily make sure $\forall 1 \leq k \leq K, f_k(\boldsymbol{0}) = 0$ and $c_k = f_k(\boldsymbol{a}_i) > 0$ as long as $\boldsymbol{a}_i \neq 0$ due to the weight positivity in PICNN. Thus, $\tilde{\boldsymbol{a}}_0 \in \tilde{\mathcal{A}}$. Consequently, we can utilize the gradient-based method to iteratively search the global optimal solution from $\tilde{\boldsymbol{a}}_0$ to the boundary point in $\partial\tilde{\mathcal{A}}$, as shown in the bottom right part of Fig. 1. Finally, the above optimization is solved multiple times, and each time we input one offline state-action pair to the second optimization in equation 3. Thus, the inference procedure brings optimal query actions.

## 5 EXPERIMENTS

### 5.1 SETTINGS

**Datasets**. We use the following datasets for testing. (1) **Grid world**. We simulate a $20 \times 20$ grid and let the agent search for the correct path to the target grid. Further, in the offline setting, we restrict the bottom right region of the grids to unseen states that the agent never explores, shown as the black region in Fig. 2a. Subsequently, in the online scenario, the start state $\tilde{s}_1$ lies in one of the unseen grids. PGO$^2$ then helps to generate action queries that enable the agent to quickly move to the target grid. (2) **Maze2D** (Fu et al., 2020). Similarly, Maze2D is an environment that requires the agent to navigate to the target location. We conduct the same treatment as **grid world** to restrict the unexplored areas in the offline dataset. Then, the agent starts at the unexplored location in the online phase. (3) **Gym domain** (Fujimoto et al., 2019). The data is gathered from OpenAI Gym benchmark tasks (Walker environments). We randomly delete $20\%$ of the historical benchmark data and use the state in the deleted dataset as the start state for online testing. (4) **Atari games** (Bellemare et al., 2013). We evaluate the performance of the agent for the image-based Atari game (Breakout). We conduct the same treatment as **gym domain** for the offline data and the online state initialization. (5) **Electric system** (Li et al., 2022). Electric systems transmit electricity from generators to loads. Proper controls are essential to guarantee system stability. Offline RL provides a chance to learn a sub-optimal policy with changing system states. We conduct the simulation for the 200-node system (Engineering Texas A&M University, 2016) based on a set of system states. For the online testing, we start at an out-of-sample state to evaluate the learned policy.

**Benchmark methods**. We introduce the following benchmark methods for comparisons. (1) **Q-Learning with Bootstrapping Error Accumulation Reduction** (Kumar et al., 2019) (BEAR). BEAR reduces the bootstrapping error and enables stable training that is robust to data distributional shifts. (2) **Conservative Q-Learning (CQL)** (Kumar et al., 2020). CQL estimates a conservative Q value that is a lower bound of the true value. Thus, CQL prevents the overestimation of the Q values and increases the generalizability of the learned policy to online implementation. (3) **Pseudometric Learning Offline RL (PLOFF)** (Dadashi et al., 2021). PLOFF proposes to learn pseudometric with the historical dataset. Then, the pseudometric works as a bonus term to restrict the state-action pairs to always stay close to the support of the historical data. (4) **Offline-to-Online RL with Balanced Experienced Replay and Pessimistic Q-Ensemble (O$^2$RL)** (Lee et al., 2022). O$^2$RL reweights the offline and online samples and utilizes Q-value ensemble to reduce overfitting.

**Metric**. We utilize the normalized score (Fu et al., 2020) between $0$ and $100$ to evaluate the performance of different methods. The normalization facilitates the comparison across tasks. The normalized score is evaluated in the online implementation after the offline training and online querying. Specifically, we consider $M \in \{20, 40, 60, 80, 100\}$ query actions in the query phase. In addition, we report the average results over $4$ seeds to guarantee a robust performance.

## 5.2 PSEUDOMETRIC GUARANTEES OPTIMAL QUERY ACTIONS

In this subsection, we use the grid world data to demonstrate the effectiveness of the learned pseudometric for online queries. Specifically, we need to validate the optimality of the query action for an unseen grid in the online phase. Therefore, we plot the query actions for 3 different methods in Fig. 2. The rest 2 methods have similar performance to the CQL method. With multiple trials, the color of the grid implies the frequency of states in the online testing.

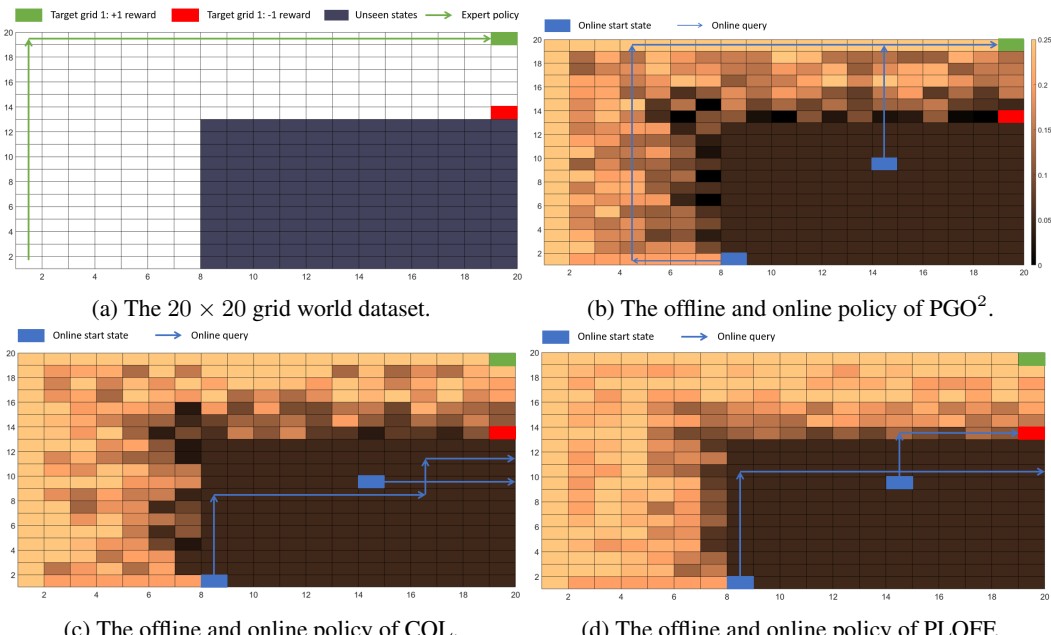

(a) The $20 \times 20$ grid world dataset.

(b) The offline and online policy of PGO$^2$.

(c) The offline and online policy of CQL.

(d) The offline and online policy of PLOFF.

Figure 2: Results of 3 different methods for the grid world dataset.

The results imply that starting from an unseen state, the proposed PGO$^2$ can successfully find the target grid (i.e., the green block) while CQL and PLOFF cannot. The key reason is that PGO$^2$ can produce the optimal query action based on the comparison to the states in the historical dataset. For the first start state $(8, 0)$, the query mechanism in PGO$^2$ will find the closest state $(7, 0)$ and infer the query action "left" that is the optimal action for $(7, 0)$. This action helps the agent to go out of the unexplored region and find the known path to the target grid. Similarly, when the start state is $(14, 9)$, PGO$^2$ will find the state-action pair, $(14, 13)$ and "up", in the historical dataset, which guides the agent to the target grid. However, for CQL and PLOFF, we directly apply the offline trained policy network to find the action, which eventually fails due to the overfitting of the neural network-based approximator.

## 5.3 OPTIMAL QUERIES AND FAST UPDATES BRING THE BEST ONLINE POLICY

In this subsection, we conduct comprehensive tests for different tasks and methods. Specifically, we restrict $M = 60$ and report the performance of the online policy obtained after the offline training and the online query. Fig. 3 illustrates the results of the normalized scores. Generally, PGO$^2$ has the best performance in the online environment. For grid world data, Section 5.2 shows that PGO$^2$ can find correct query actions based on the most similar state-action pairs in the historical dataset. For other methods, however, their agents will get stuck in the unexplored region. For more complex tasks, PGO$^2$ is still better since it can efficiently use online information. Finally, O$^2$RL also has the mechanism to largely make use of the online query results. However, since our query number $M$ is limited compared to the historical dataset, O$^2$RL does not have significant improvement.

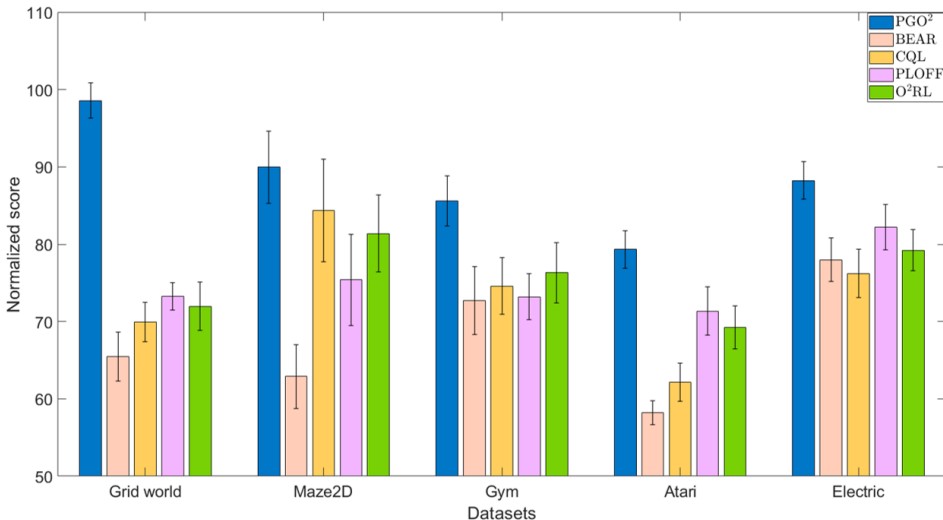

Figure 3: Results of different methods for online tests in different datasets.

## 5.4 SENSITIVITY ANALYSIS: A SMALL NUMBER OF ONLINE QUERIES IS ENOUGH

We further study the problem of how many queries are needed to produce a high-level online policy. Thus, we vary $M$ and test different datasets. Fig. 4 demonstrates the results. We have the following observations. First, as $M$ increases, PGO$^2$ and O$^2$RL have significant improvements while the rest methods only have slight improvements. This is because PGO$^2$ and O$^2$RL have effective mechanisms to update the policy based on the online data. However, the improvement for PGO$^2$ is much higher since PGO$^2$ can better use the online queries results based on our unique designs of parameter-sharing between the Q-network and the pseudometic learning network. Second, when $M$ is small, PGO$^2$ still brings relatively large scores, which further validates the model's effectiveness.

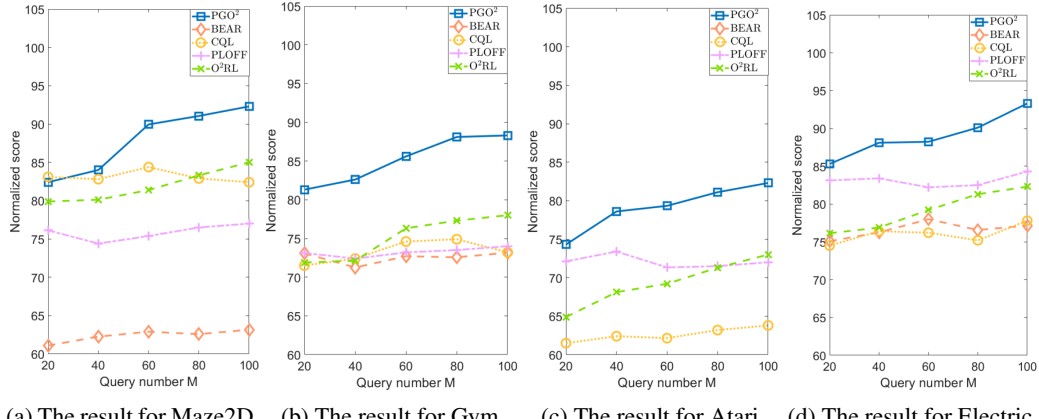

(a) The result for Maze2D.  (b) The result for Gym.  (c) The result for Atari.  (d) The result for Electric.

Figure 4: Results of sensitivity analysis for different datasets.

## 5.5 COMPARISONS TO PRE-TRAINING-BASED OFFLINE TO ONLINE RL

In this subsection, we treat the pure offline RL models as pre-training for online implementations. Then, we continue the model updates in the online phase using query results. We fix $M = 100$ and the results are shown in Table 1. Although different models have the post-training process after the online results, our PGO$^2$ still achieves the best performance due to its efficient utilization of online query results.

Table 1: Results of different methods for offline pre-training and online post-training.

| DATASET | PGO$^2$ | BEAR | CQL | PLOFF | O$^2$RL |
|---------|---------|------|-----|-------|---------|
| MAZE2D | 92.3 | 66.4 | 85.2 | 79.7 | 85.2 |
| GYM | 88.2 | 72.3 | 73.5 | 73.8 | 78.9 |
| ATARI | 83.2 | 75.6 | 67.8 | 74.5 | 73.8 |
| ELECTRIC | 94.2 | 78.6 | 76.2 | 88.6 | 82.3 |

## 6  CONCLUSION AND FUTURE WORK

We propose PGO$^2$: Pseudometric Guided Offline-to-Online Reinforcement Learning. PGO$^2$ is effective with (1) optimal online action queries and (2) efficient online Q-network updates. We demonstrate that the effectiveness comes from the coupling between pseudometric learning and policy network learning. In addition, we enforce input convexity to the framework to guarantee query optimality. Finally, we test different datasets to demonstrate the superiority of PGO$^2$. Possible future directions may include (1) the development of evaluations on the transferability from offline to online after the online query and (2) model improvement in the face of sparse rewards.

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
