# OpenReview forum: "Pseudometric guided online query and update for offline reinforcement learning"
_ICLR.cc/2023/Conference — Submitted to ICLR 2023_

### Official Review · Reviewer_pNvF · 2022-10-25

**Confidence:** 4
**Correctness:** 3
**Technical Novelty And Significance:** 3
**Empirical Novelty And Significance:** 3
**Recommendation:** 6

**Clarity, Quality, Novelty And Reproducibility:**

My main issue with this paper is in its clarity, as detailed above. The algorithm builds on PLOFF from [Dadashi et al., 2021], but it does appear to be novel.

There was no code provided with the submission, nor was there a clear and extensive description for reproducibility included in the paper (there was also no appendix). Algorithm 1 and 2 outline the algorithm, but there are a number of notational issues in Algorithm 1 (detailed above), which bring into question the facility to reproduce results.

**Strength And Weaknesses:**

# Strengths
This paper presents an interesting algorithm that builds on the work of [Dadashi et al., 2021] for more efficient online environment interactions when training from an offline dataset.

The use of pseudometrics is well motivated and reasonably well evaluated.

# Weaknesses
The main weakness of this paper is in the clarity of presentation, outlined below.

## Algorithmic issues
1. Why is the method restricted to a single start state $\tilde{s}_1$? What about an initial state distribution?
1. Figure 1 helps calrify the method, but it is not clear where $\Psi$ fits in this figure.
1. The idea of coupling metric and Q-value learning in a parameter-sharing manner has already been done in [MICo](https://papers.nips.cc/paper/2021/hash/fd06b8ea02fe5b1c2496fe1700e9d16c-Abstract.html), so probably worth citing.
1. In the second paragraph of page 4 it says "$\Phi(\cdot)$ considers a uniform distribution over the action space." but this is not entirely correct, since in (2) the actions are sampled from $\mathcal{D}$ (replay buffer) when computing $\mathcal{L}_{\Phi}$.
1. One point that is not clear to me is how critical the use of PICNN is to this method. While I can appreciate the desire to have a convex optimization problem, it is not entirely clear to me how critical this choice is to the performance of the algorithm.
1. In the last paragraph of page 4 it proposes using the projected Newton method to find $\max_a$. If $\mathcal{A}$ is small, couldn't you just do a simple iteration over all actions?
1. In the last line of page 4, why does using a bundle of PICNNs "guarantee to find the optimal query action", if they're all outputting different estimates for the same $a$?
1. In the first paragraph of section 3.3 it says "one cannot directly infer the query action by $Q(\cdot)$", but one can, in principle; if $Q$ generalizes well it might be reasonable.


## Notational issues
1. In the second paragraph of section 3.1, the term inside the expectation is wrong, as it is saying $s_k, a_k$ must be equal to the original $(s,a)$ for all $k$. This should only be the case for $k=0$.
1. The maximization in the next line says $\max_{\pi}Q^{\pi}(s, a)$, but it should be quantified to be for all $s$, or at least for all $s$ from some initial state distribution.
1. In the "Query Task" bullet on page 3, are the optimal query actions an open-loop policy? The way it's written it appears to be the case, which seems strange.
1. In the second line of equation (2) it is written $\|\| \Psi(s_i) - \Phi(s_j) \|\|$, but I believe this should be $\|\| \Psi(s_i) - \Psi(s_j) \|\|$.
1. In Algorithm 1 in the innermost for loop, shouldn't the parameters of $Q'$ be $s_{m+1}$ and $a_{m+1}$ (instead of $m$)? Similarly, in the line below it shouldn't it be using $s_m,a_m$ (instead of $m+1$)?
1. In the first paragraph of section 3.3 the phrase "learning $d_\Phi$ has good convergence to the unique fixed point pseudometric to measure the bisimulation of state-action pairs in MDPs." does not make much sense. Is the fixed point the bisimulation metric?
1. A related point: it is assumed bisimulation is a "reasonable" thing to use, but bisimulation has not really been introduced properly at this point, so it's not clear _why_ it's reasonable.
1. In  equation (3) is the $\arg\min$ only over $s_c$? It's not clear.
1. In the last line of the derivation in proof of Proposition 2 in the $(f_k(\tilde{\mathbf{a}}) - c_k)$ term it seems like there's a $t\mathbf{h}$ term missing inside the $f_k$.
1. In the last line of the proof of Prop 2, it says $L(\tilde{\mathbf{a}} + t\mathbf{h})=\frac{d}{dt}$, ... what does that RHS mean? It appears to be incorrect o a typo.

## Empirical evaluation
1. In section 5.2, what does "the probability of the state in the implementations" mean? Probability of what?
1. How stable are the trajectories in Figure 2? It would be good to superimpose multiple policy rollouts to evaluate this.
1. Since the values in Figure 3 are normalized, you should use more robust metrics such as IQM, as presented in [Deep Reinforcement Learning at the Edge of the Statistical Precipice](https://proceedings.neurips.cc/paper/2021/hash/f514cec81cb148559cf475e7426eed5e-Abstract.html) (and [corresponding library](https://github.com/google-research/rliable).
1. Continuing a point made above, I think it would be good to have an ablation study where PICNN is not used (e.g. convexity is not guaranteed). Although perhaps not theoretically principled, this may yield better empirical performance.


# Questions / suggestions
1. In the introduction it would be good to discuss connections with the findings from:
    * [Georg Ostrovski, Pablo Samuel Castro, Will Dabney. The Difficulty of Passive Learning in Deep Reinforcement Learning. NeurIPS 2021](https://openreview.net/forum?id=nPHA8fGicZk)
1. In the second paragraph of page 2 it says "must have a small difference between the expected rewards", but this is not accurate. It is the difference between the sum of discounted returns.
1. In the **State-action Similarity Metric for MDPs** paragraph in section 2, you should also reference
    * [Jonathan Taylor, Doina Precup, Prakash Panagaden. Bounding Performance Loss in Approximate MDP Homomorphisms. NeurIPS 2008.](https://proceedings.neurips.cc/paper/2008/hash/6602294be910b1e3c4571bd98c4d5484-Abstract.html)
    * [Pablo Samuel Castro, Tyler Kastner, Prakash Panangaden, Mark Rowland. MICo: Improved representations via sampling-based state similarity for Markov decision processes. NeurIPS 2021](https://papers.nips.cc/paper/2021/hash/fd06b8ea02fe5b1c2496fe1700e9d16c-Abstract.html)
1. In the second line of equation (2), could you have independent action samples for $s_i$ and $s_j$, instead of the same $u_k$ for both?
1. In the last paragraph of page 4 what is meant by "action-Q pairs", and what is meant by "helps to do maximization by comparison"?
1. In the last paragraph of page 4 what is meant by "an inference problem for PICNN"?
1. In the last line of page 4, why is the output considered negative (e.g. $-\Phi(s,a)$)?
1. In Algorithm 1, wouldn't $\gamma$ and $T$ be inputs, rather than values to initialize?
1. In Algorithm 1, does $\mathcal{B}_1$ start empty?
1. Nit: In Algorithm 1 ity says "Solving the optimization in equation 2", but it's not actually _solving_ at each step, but approaching solution, right?
1. In the paragraph below Algorithm 1 [Fan et al., 2020] is cited for the use of a target Q-network, but this is an idea that was introduced by Mnih et al., 2015, in the original DQN paper.
1. In section 5.3, just to confirm: there is no learning happening during eval, correct?


# Minor points
1. In the second paragraph of the introduction it should say "The **central** reason is the data distributional ..."
1. In the end of the first paragraph in page 2 I think it'd read better if it said "we **require** a good pseudometric" (instead of demand).
1. In the start of section 3.2 use `\citet` to avoid having parentheses around the noun.
1. In the second line of section 3.2 you can remove "(i.e. a pseudometric)".
1. In the third line of section 3.2 there is a typo: "pesudometric"
1. In the second line of page 4 it should say "$\Phi(s, a)$ is **a** feature embedding..."
1. In the second-to-last paragraph of page 4, it should read "trained by minimizing the loss**es** in equation 2"
1. In the last paragraph of page 4 it should say "output of the Q-network **are** the action-Q pairs"
1. The sentence right before section 3.3 does not need to start with "In general"

**Summary Of The Paper:**

This paper proposes a method for offline reinforcement that relies on limited queries to the environment for dealing with out-of-distribution actions during the online learning phase. The authors use a state-action (learned) pseudometric to compare online and offline states and determine which actions are best to query. Part of this paper's method is the use of a siamese network to simultaneously update the Q-values and the pseudometric.

**Summary Of The Review:**

Overall an interesting algorithm that is well motivated. There are a number of issues around clarity and notation, but I think most of these should be fixable during rebuttal. The question around reproducibility should also be addressed.

I am on the fence with this paper, but leaning towards an accept. The final decision will be contingent on the authors' response during the discussion period and the other reviews.

---

> ### Author Response · Authors · 2022-11-19
> **Response to reviewer 3**
>
> Q1. Algorithm Issues.
>
> Response: Thanks a lot for pointing out these questions and we really appreciate the reviewer's effort to make our paper better. For your algorithmic issues, we answer them as follows. (1) We assume the initial state is observable and known. Thus, there is no need to consider an initial state distribution. (2) We update Fig. 1 and add the notation of $\Phi$. (3) We cite the paper you mentioned. (4) We agree with you and remove the statement related to $\Phi(\cdot)$. (5) First, [1] shows that Input Convex Neural Network (ICNN)-based RL brings better performances in diversified RL tasks. Second, we further extend the power of ICNN to design the convex optimization-based action generation in the query process. This guarantees a global optimal query action for each given state. Such a guarantee is powerful since the local optimal action may make the querying process trapped in low-reward regions, as shown in Fig. 2 in our paper. (6) We agree that if the action space is small, we can check all possible actions and find the best one. (7) The bundle PICNN is to estimate the features of the state-action pair, which is used to calculate the pseudometric. Namely, their different outputs together determine the distance calculation. (8) We agree that this sentence is confusing. We change from "cannot" to ``may not".
>
> Q2. Notational issues.
>
> Response: Thank you for your careful reading. For the notational issues, we make the following clarifications. (1) We change the equation to $Q\^{\pi}(\boldsymbol{s},\boldsymbol{a})=\mathbb{E}\big[\sum\_{k=0}\^{\infty}\gamma\^kr(\boldsymbol{s}\_k,\boldsymbol{a}\_k,\boldsymbol{s}\_{k+1}|\boldsymbol{s}\_0=\boldsymbol{s},\boldsymbol{a}\_0=\boldsymbol{a})\big]$. (2) We add the explanation that the optimization is to obtain the policy function for all states. (3) We add an explanation ``Note that repetitions can happen for the query action sequence." (4) We change it to $\big|\big|\Psi(\boldsymbol{s}\_i)-\Psi(\boldsymbol{s}\_j)\big|\big|$. (5) We change it to $y_m=r_m+\gamma Q^{'}(\boldsymbol{s}\_{m+1},\boldsymbol{a}\_{m+1})$. (6) Yes, $d\_{\Phi}$ can converge to a fixed point that is a pseudometric. You can refer more to \cite{dadashi2021offline}. (7)  According to \cite{dadashi2021offline}, bisimulation measures the similarity of two state-action pairs based on the reward. Next, bisimulation measure has strict theoretical foundations and good empirical estimation convergence. Thus, it's reasonable to utilize bisimulation measures for generating online queries of actions. (8) $\arg\min$ is over $\tilde{\boldsymbol{a}}$ to produce the online query action. (9) Based on the chain rule, one can insert $t=0$ in $f_k(\tilde{\boldsymbol{a}}+t\boldsymbol{h})-c_k$ since this term represents the function value (not the derivative) when $h\rightarrow 0$. (10) We delete "$=\frac{d}{dt}$".
>
> Q3. Empirical evaluations.
>
> Response: (1) It should be the frequency of online states in the online implementations. (2) Yes, we conduct multiple tests and calculate the empirical frequency in Fig. 2. (3) We follow the normalized score mentioned in [2]. (4) We appreciate this suggestion and this work will be done in future work.

---

> > ### Author Response · Authors · 2022-11-19
> > **Reference**
> >
> > [1] Amos, Brandon, Lei Xu, and J. Zico Kolter. "Input convex neural networks." International Conference on Machine Learning. PMLR, 2017.
> >
> > [2] Kumar, Aviral, et al. "Conservative q-learning for offline reinforcement learning." Advances in Neural Information Processing Systems 33 (2020): 1179-1191.

---

### Official Review · Reviewer_dmti · 2022-10-25

**Confidence:** 3
**Correctness:** 2
**Technical Novelty And Significance:** 1
**Empirical Novelty And Significance:** 1
**Recommendation:** 3

**Clarity, Quality, Novelty And Reproducibility:**

Clarity:

- Poor. The paper is hard to read and follow, though the idea is simple. It does not do a good job to share the main message.

Quality and Novelty:

- This paper has very limited novelty compared with the original paper proposing pseudo-metric for measuring "support distance". The utilization of the pseudo-metric in the online setup is kind of counter-intuitive without any explanation.

- The empirical results are also very limited, with weak baselines.



**Strength And Weaknesses:**

Strength:

- The setup this paper studies is kind of interesting. There is some statistical limit of learning from purely offline dataset, and sometimes online query under some costs is available. The setting is of practical relevance and should be relevant to various settings.

- Empirical results show better performance compared with purely offline setting, though straightforward.

Weakness:

- This paper is very difficult to read, as it does not provide a clear motivation of the method, and why the pseudo-metric would be helpful for online exploration stage. The main sections is a mix of methods, algorithms and implementation details, which makes it difficult to follow.

- The novelty is limited, as the notion of using this type of pseudo metric is proposed in [1], and this paper utilizes it the same way as the original paper in the offline stage. The use of it in the online stage seems to help query the new data? However, it is super unclear to me why we want to query nearby data points? It is very different from most online exploration literature. There is no justification here.

- The empirical results are limited, most of the algorithms are purely offline setting, there are some works such as utilizing offline RL dataset for pre-training. Could the authors add more comparable baselines?


Ref:
[1]. Robert Dadashi, Shideh Rezaeifar, Nino Vieillard, Le ́onard Hussenot, Olivier Pietquin, and Matthieu Geist. Offline reinforcement learning with pseudometric learning. In International Conference on Machine Learning, pp. 2307–2318. PMLR, 2021.

**Summary Of The Paper:**

This paper studies the offline RL with online query setting, i.e., given a dataset and online access to the environment, how to effectively learn the optimal policy. It proposes Pseudometric Guided Offline-to-Online RL (PGO2). The main idea is to learn a pseudo metric that measures the closeness from the support of the training data in the offline training stage, and using this metric to query nearby actions in the online setting to gather more data. Experiments on different datasets show better performance compared with purely offline algorithms.

**Summary Of The Review:**

The current version of the paper is a clear reject due to:

- The low clarity in the paper.

- The limited novelty in the method, and no theoretical justification of the online exploration part is given.

- The weak baselines for the empirical setting.

---

> ### Author Response · Authors · 2022-11-19
> **Response to reviewer 2**
>
> Q1. Paper clearity.
>
> Response: Thanks for your comments. We improve the description of the motivation and explanation. Further, we reorganize the paper and move the implementation details and proofs to the appendix to make the presentation clearer.
>
> Q2. Motivations and Novelty.
>
> Response: Thank you for your comments. We make the following claims to elaborate on our motivations and novelty. First, our paper focuses on the scenario when we have a large set of offline data, and several chances to conduct online queries. Then, we need a systematic design to guide the online query process to obtain proper online actions and improve the offline model based on the query results. Since the query chances are limited and valuable, we want the query-based mechanism to (1) fully utilize the learned model from the offline dataset and (2) efficiently update the learned model based on limited query results. For (1), we need to generate as close state-action pairs as possible to the offline data. Then, we can verify if the learned model is suitable for the online environment. For (2), we need to update a large number of parameters using limited data. Thus, the problem is in general nontrivial.
>
> Second, to tackle the above problem, we propose a novel design based on pseudometric learning [1]. However, we make several unique and significant contributions. (i) We develop a parameter-sharing structure in the Siamese network for pseudometric learning. This enables a sufficient update with limited online query results. More specifically, we compare each query result with respect to the complete historical dataset in the Siamese network, leading to sufficient evaluations of the query results and parameter updates for the policy network and handling the above goal (2). (ii) We further develop a convex optimization to obtain query actions, based on the convex design of our policy network and pseudometric framework and proper initialization. This helps us to obtain the global optimal actions for online queries for the above goal (1).
>
> Q3. Empirical Evaluations.
>
> Response: First, we utilize Offline-to-Online RL with Balanced Experienced Replay and Pessimistic Q-Ensemble ($\text{O}^2$RL) [2] as comparisons, which is the state-of-the-art method that bridges offline and online RL training.  Second, we also consider the work utilizing offline RL as pre-training and add Table 1 in the revised version. Please check the revised paper.

---

> > ### Author Response · Authors · 2022-11-19
> > **Reference**
> >
> > [1] Dadashi, Robert, et al. "Offline reinforcement learning with pseudometric learning." International Conference on Machine Learning. PMLR, 2021.
> >
> > [2] Lee, Seunghyun, et al. "Offline-to-online reinforcement learning via balanced replay and pessimistic q-ensemble." Conference on Robot Learning. PMLR, 2022.

---

### Official Review · Reviewer_Ggyx · 2022-10-25

**Confidence:** 4
**Correctness:** 3
**Technical Novelty And Significance:** 2
**Empirical Novelty And Significance:** 3
**Recommendation:** 5

**Clarity, Quality, Novelty And Reproducibility:**

Clarity: Not everything is clear, see my comments above.

Quality: The quality of the work is good, but it could be improved with more realistic experiments and thorough ablations.

Novelty: There are several existing methods dealing with similar problems, but the proposed method seems to be combining existing components in a novel way.

Reproducibility: Not all the details are discussed to insure reproducibility, for example, information about the offline datasets is missing. The method seems to be quite complex and thus code would be required to ensure reproducibility.


**Strength And Weaknesses:**

Strengths:
- The problem of adapting offline policies with the small amount of online interactions is certainly interesting and deserves attention.
- The authors test their method on multiple diverse domains.
- Several diverse baselines are used in the comparison.
- Experimental results include experiments for understanding qualitative behavior of the agent.
- Experimental results show the sensitivity of the method to the number of online trajectories.

Weaknessess:
- It seems that the paper uses an assumption that part of the initial states are hidden from the agent during offline training. While I understand that a problem of such distribution shift can occur naturally due to the fact that the dataset is limited, the experimental settings are such that part of the states are manually deleted from the offline training. I find this setting somehow artificial. It would be more interesting and convincing if the method would demonstrate its advantages in the more realistic settings in addition to the manually constructed test case. Moreover, this assumption should be stated clearly at the beginning of the paper, currently it becomes clear only when reading about the results.
- It is not clear from the paper what kind of offline dataset is required for training. From section 3.2 it sounds that the dataset should contain uniformly random actions. Is this the case? How realistic is it to collect such a dataset in practice? How would the method perform if this assumption is violated? How was the dataset created? Is the data part of an existing offline RL benchmark? Also, one of the claims about the proposed method is that it does not require as much offline data as other methods but no information about the sizes of the offline datasets are provided.
- I found some parts of the paper not clear and the writing needs a bit more work. For example, on the first page "seeks sub-optimiality" (do you mean optimality?), "centric reason" (central reason?), "query reasonable actions" (what does it mean to query an action?). In terms of mathematical details, I find some parts confusing. For example, r: S x A, but then r(s_k, a_k, s_{k+1}), \Phi function in equation 2 sometimes takes s and a and sometimes only s. The figures in the experiments do not have sufficient explanations, for example, while figure 2 looks interesting, it is not clear to me what color is representing (the text says "probability of the state in the implementations", what is implementation? Is it during online fine-tuning? Or are states sampled during offline training? Or is it during the deployment?) etc.
- The proposed method is quite complex and includes many moving parts. It would be beneficial for the reader to understand how various components contribute to the performance of the method if introduced independently. Usually in offline RL, applying the policies from online RL directly results in very poor performance. Would this also be the case with the proposed method if no online updates are done? Or is this problem mitigated with the elements of representation learning? Also, how were the different hyperparameters set? For example, frequencies of updates of various parts of the network.
- The proposed method with pseudometric learning sounds like it is from a group of representation learning methods in RL. Related work should include works on it. Maybe even a representation learning baseline could be beneficial.
- Limitations of the method are not discussed.

Questions and comments
- Some part of the method deals with finding the maximum of Q(s, a) where a is the input to the network. Wouldn't methods like DDPG already address such a problem?
- I think the trick with having a "target" Q network is much older than 2020 and was introduced as part of the DQN method.
- Theoretical analysis does not need to include the proposition from the existing work if it is not required for other proofs.


**Summary Of The Paper:**

This paper proposes a new method for learning policies first from offline dataset and then from a limited number of interactions with the real environment. The special feature of the method is that it learns a pseudometric that corresponds to the similarity between states in the environment. Then, when during online interactions the agent faces some unseen initial states, it selects the action by retrieving the most similar state in the existing offline dataset. The experimental results demonstrate promising behavior in several environments in the situations when part of the initial state space is hidden.

**Summary Of The Review:**

I am leaning toward the rejection of the paper mainly because I find the setting and its motivation (missing part of the states space) not very convincing given the nature of the experiments where states were manually deleted. The nature of the offline datasets that are required for training should be clarified as well. Besides, I think the clarity of the paper could be improved and experiments could be done more thoughtfully to include ablations of the components of the method.

---

> ### Author Response · Authors · 2022-11-19
> **Response to reviewer 1**
>
> Thanks a lot for all of your comments. We provide thorough responses as follows.
>
> Q1. Paper assumptions.
>
> Response: Thanks a lot for your comments. In the paper, we manually delete some states to make sure the online start state has a certain distance from the offline dataset. Such a design is only for empirical evaluation in extreme tests and our method generally produces a reasonable and optimal action for arbitrary states, no matter whether the online initial state exists or not in the historical dataset. This is because in our query part in Equation (3), if the online initial state exists in the historical data, the query process will produce the optimal action based on the offline RL.
>
> Q2. Random actions in offline datasets.
>
> Response: We don't need uniform actions to appear in the historical dataset. We follow [1] to achieve pseudometric learning and employ their formulas as Equation (2) in our paper. The uniformly distributed actions $\boldsymbol{u}_k$ are only for evaluations to calculate the value function $\Phi(\boldsymbol{s}\_i,\boldsymbol{u}\_k)$ in Equation (2). However, the historical datasets will provide $(\boldsymbol{s}_i,\boldsymbol{a}\_i,r\_i,\boldsymbol{s}\_{i+1})\sim \mathcal{D}$, $(\boldsymbol{s}\_j,\boldsymbol{a}\_j,r\_j,\boldsymbol{s}\_{j+1})\sim \mathcal{D}$, where $\boldsymbol{a}_i$ and $\boldsymbol{a}_j$ are true actions in the historical dataset and they are not uniformly distributed. More details of the formula can be checked in [1].
>
> Q3. Unclear parts in the paper.
>
> Response: Thanks a lot for your comments. We have modified the writing based on your suggestions. Second, for "query an action", we change it to ``produce a query action to the environment". Third, we modify the definition of the reward function to $r:\mathcal{S}\times \mathcal{A}\times \mathcal{S}\rightarrow \mathbb{R}$. Fourth, for Equation (2), please note $\Phi(\cdot)$ and $\Psi(\cdot)$ are different functions that evaluate the sate-action pair and the state, respectively. Fifth, the color in Fig. 2 represents the frequency of states in the online testing.
>
> Q4. Tests without online queries.
>
> Response: Thanks for these questions. We currently don't include the tests when only the offline dataset is used. This is because we target utilizing online queries to significantly improve the offline model. However, we agree that purely comparing the offline performance is another perspective to evaluate the model. We will add them in future work.
>
> Q6. More literature and limitation discussion.
>
> Response: Thank you for brining the related topic to us. We will add the review and limitation discussion in the final version.

---

> > ### Author Response · Authors · 2022-11-19
> > **Reference**
> >
> > [1] Dadashi, Robert, et al. "Offline reinforcement learning with pseudometric learning." International Conference on Machine Learning. PMLR, 2021.

---

> > ### Comment · Reviewer_Ggyx · 2022-11-29
> > **Thank you for the comments**
> >
> > I would like to thank the authors for responding to my questions and revising the manuscript.
> > However, I would like to keep my original rating and I am leaning towards the rejection of the paper. I think the responses that the authors provided should be part of the experimental studies and motivation (for example, if the method is beneficial without removing parts of the state space, this should be shown in an experiment). Besides, I think the paper would still benefit from more thorough text revision as other authors expressed the same concerns about the clarity of the paper.

---

### Decision · Program_Chairs · 2023-01-20

**Decision:**

Reject

**Justification For Why Not Higher Score:**

The paper's novelty is limited and the presentation is not clear enough.  There is also a disconnection between the setting/motivation and the experiments where states are deleted.  Although the rebuttals have alleviated some concerns, the paper is not yet ready and needs a new round a revision and review.

**Justification For Why Not Lower Score:**

N/A

**Metareview: Summary, Strengths And Weaknesses:**

This paper tackles offline-to-online reinforcement learning by adapting the offline learned policy via a learned pseudometric, which accounts for the similarity between online and offline states, allowing optimal query actions to be inferred subject to limited chances.  A Siamese network is employed to simultaneously update the Q-values and the pseudometric.

The paper addresses an important problem, and the experiments have covered multiple domains and baselines, with some promising results.  The reviewers have raised concerns on novelty, clarify in presentation, and the disconnection between the setting/motivation and the experiments where states are deleted.  Although the rebuttals have alleviated some concerns, the paper is not yet ready and needs a new round a revision and review.